# *CHAMPP4KIDS*: Mixed methods study protocol to evaluate acceptability and feasibility of Parenting for Lifelong Health materials in a Canadian context

**Andrea Gonzalez**[1]*, **Susan M. Jack**[2], **Amanda Sim**[1], **Jenna Ratcliffe**[1], **Mari Dumbaugh**[3], **Teresa Bennett**[1], **Harriet L. MacMillan**[1,4]

1 Department of Psychiatry and Behavioural Neurosciences, McMaster University, Hamilton, Ontario, Canada, 2 School of Nursing, McMaster University, Hamilton, Ontario, Canada, 3 School of Public Health, University of Illinois-Chicago, Chicago, Illinois, United States of America, 4 Department of Pediatrics, McMaster University, Hamilton, Ontario, Canada

* gonzal@mcmaster.ca

**Data Availability Statement:** No datasets were generated or analysed during the current study. All

## Abstract

### Background

Parents and caregivers play a key role in children's healthy development and well-being. Traditional parenting interventions promote positive parenting practices and are key to preventing child maltreatment. However, numerous barriers can limit access to programs, barriers which were further exacerbated by the COVID-19 pandemic. The Parenting for Lifelong Health group developed mass media and public health communication materials to promote positive caregiving behaviours on a population level. The Champions of Positive Parenting 4 Kids (CHAMPP4KIDS) study will examine the acceptability and feasibility of these materials for service providers and caregivers of children aged 2–6 years in Ontario, Canada.

### Methods

This study will use a convergent mixed-methods design. Consenting service providers (n = 200) and caregivers (n = 100) will complete a quantitative survey to rate, rank and give feedback on Parenting for Lifelong Health tip sheets and social media ads. Caregivers will also complete self-report scales measuring depression and anxiety. We will hold focus group discussions with a sub-sample of surveyed providers (n = 40) and caregivers (n = 25). An adapted Trials of Improved Practices methodology will explore caregiver perspectives after implementing the tip sheets. Primary quantitative outcomes will be descriptive statistics of rankings, Likert Scale scores and descriptive analysis of caregiver depression and anxiety. Qualitative data will be analyzed using Rapid Qualitative Inquiry and triangulated through a convergent coding matrix.

relevant data from this study will be made available upon study completion.

**Funding:** Public Health Agency of Canada awarded to AG, HM, SJ, TB. AG supported by a Tier II Canada Research Chair in Preventive Interventions and Family Health. HM is supported by the Chedoke Health Chair in Child Psychiatry. The funders did not and will not have a role in study design, data collection and analysis, decision to publish, or preparation of the manuscript.

**Competing interests:** The authors have declared that no competing interests exist.

**Abbreviations:** CFIRC, Consolidated Framework for Implementation Research Constructs; CHAMMP4KIDS, Champions of Positive Parenting 4 Kids; DOI, Rogers' Diffusion of Innovation Theory; FGD, Focus group discussions; HiREB, Hamilton Integrated Research Ethics Board; ICF, Informed consent form; PLH, Parenting for Lifelong Health; RQI, Rapid qualitative inquiry; TIPS, Trials of improved practices; WHO, World Health Organization.

## Discussion

The Parenting for Lifelong Health COVID-19 parenting materials offer succinct, engaging parenting information in a mass media format that addresses some challenges associated with accessing in-person programming. The CHAMPP4KIDS study will provide mixed methods insights on the materials' acceptability and feasibility from different groups in a Canadian context, with a focus on marginalized families. The use of Trials of Improved Practices methodology could prove a useful tool for participant-led adaptation of existing parenting, early childhood development and other health intervention materials.

## Background

Parents and other caregivers play a key role in children's healthy development and well-being [1–4]. Traditional parenting interventions, which can be delivered at home, in community settings or online to individual parents, families or groups, promote positive parenting practices and safe, stable, and nurturing relationships between caregivers and children, and are key to preventing child maltreatment [1, 2, 5–14]. For example, parenting programs can reduce risk factors for child maltreatment such as harsh, punitive, or unpredictable parenting practices and poorly managed and conflicted households and improve emotional and behavioural adjustment of children, and enhance the psychosocial well-being of caregivers [6, 7, 15–18]. In 2023, the World Health Organization (WHO) developed new guidelines on parenting to prevent child maltreatment, improve parent-child relationships and promote positive development in children and youth [14].

Program implementers have faced challenges engaging caregivers as reflected by low program attendance and low completion rates due to factors such as high demands of programs on caregivers' time, scheduling conflicts, logistics such as transport and child care, fatigue, and lack of motivation [18, 19]. Even if programs are free, indirect costs and other barriers prevent families, especially those facing socioeconomic challenges, from participating [18]. In addition, many programs are targeted or only offer services to caregivers of children who are screened 'at-risk' and qualify for services [20]. Stigma and numerous barriers to accessing screening and services can also limit program reach [21, 22]. Research indicates that caregivers and families who live in rural areas, are racialized, Indigenous, low income, single mothers or migrants face additional layers of barriers to accessing parenting programming and other child mental health services [18, 23–28]. All of these factors limit the potential impact of interventions for families most in need of services, in addition to the population at scale [29].

To overcome the challenges of engaging caregivers in these traditional parenting programs, and extend the reach and benefits of evidence-based interventions to more families, universal programs engage both families at risk of developmental difficulties and those in the general population [22, 29]. Though more limited than the evidence base on targeted parenting programs, research does demonstrate that universal, population-based approaches to parenting programs may address the limited reach of programs focused exclusively on engaging individual caregivers and households [13, 21].

However, it is important to note that population-level programs can increase inequities if relative disadvantage between families is left unaddressed [21]. If policies, services, and programs include a range of responses to account for different levels of need, disadvantage, and barriers within the population, a blend of both universal and more targeted approaches to parenting programs can offer support proportionate to families' needs [30].

Combining mass media and public health communication is one population-level approach to promoting positive caregiving behaviours, while addressing the challenges facing traditional parenting programs. Public health approaches to behaviour change often involve media as a vehicle for influencing knowledge, attitudes, norms and behaviours [31]. Mass media public health campaigns can be delivered via various formats including print, television, radio, social media, and other digital/online platforms [32]. Positive parenting campaigns using mass media would complement, not replace, existing services by extending their reach to populations who often face barriers to connecting directly with social services and healthcare organizations. Mass media campaigns could offer a more accessible, efficient and affordable format for providing quality information about positive parenting strategies, without requiring caregivers to spend time and resources physically attending services to receive the same information [18]. Even modest effects of a program with wide reach can translate into substantial, meaningful benefits to society when multiplied across individuals and households [18].

While some research shows the positive effects of mass media on child health behaviours [32] and parent-adolescent communication [33], there is little research examining mass media campaigns designed to enhance parenting and improve child outcomes [1, 32, 34]. Other mass media studies demonstrating positive results were only conducted with infants, or are confounded by varying methodology such as combining home visits and group programs with mass media approaches [18, 35, 36].

In addition to the common barriers noted above, the COVID-19 pandemic, and social protection strategies implemented to mitigate the spread of the virus, created additional barriers to service provision and families' access to community supports. The COVID-19 pandemic disrupted adult and child functioning worldwide as households and individuals were confronted with, and continue to experience, the effects of illness, financial insecurity, isolation, school closures, increased alcohol use and mental health problems, and reduced access to healthcare and social services [37–45], and it is widely known that COVID-19-related outcomes were more widely and severely experienced by racialized, Indigenous, rural, migrants and low income individuals and communities [46–52]. Community surveys show that pandemic-related stressors are also associated with decreased parenting quality [37, 38, 53–59]. The intersection between existing structural disparities between populations, a heightened need for assistance during the pandemic and a decrease in access to services and community supports is a paradox that reveals the need for a radical change in our approach to providing family assistance during public health emergencies and beyond [41, 44, 59].

In response to the challenges the COVID-19 pandemic presented for children, caregivers and families around the world, the Parenting for Lifelong Health (PLH) group, in collaboration with the WHO and UNICEF, developed open-source evidence-based playful parenting resources from their existing parenting programs to reach caregivers at the population level [5, 60, 61]. Available in over 100 languages, these resources aim to impart knowledge and provide support and concrete tools to caregivers which, in turn, assist children in reaching their full developmental potential [5, 62]. The PLH resources are diverse in format; they include parenting tip sheets formatted as digital infographics, audio packs, social media kits and public service announcements. These resources are available for caregivers, providers, educators and the public via the PLH web page (covid19parenting.com) [61]. While these materials were especially pertinent during the height of the pandemic, evidence demonstrates that the need for parenting support extends beyond the time of widespread service restrictions related to COVID-19 [21, 29]. Gaps and inequities in parenting program service delivery and uptake that were already existing pre-pandemic have likely widened due to stressors experienced by caregivers, especially in marginalized families [41].

The PLH COVID-19 parenting tip sheets and social media ads are two resources for care-givers of children two to six years old included in the open-source PLH package [62]. The tip sheets provide caregivers with condensed information on concrete strategies to build positive relationships, divert and manage challenging child behaviours, and cope with parenting stress. Each tip sheet focuses on a different aspect of caregiving and offers information and strategies for children of differing ages and stages of development, from babies to teenagers. Examples of tip sheets topics include *One on One Time; Keeping Calm and Managing Stress; Learning through Play*; and *When Children Misbehave* (https://www.covid19parenting.com/#/tips#). The social media kits include evidence-based parenting tips formatted for sharing on social media platforms (https://www.covid19parenting.com/#/socialmedia#).

As of January 2023, PLH COVID-19 playful parenting resources had reached over 210 mil-lion people globally, mostly in Asia and Africa [63]. Only one study to date has examined the feasibility and acceptability of the PLH parenting tip sheets across six middle- and low-income countries [5]. Caregivers in the study said they gained important information and practical strategies from the tip sheets to support them during the pandemic, including reductions in their use of harsh discipline [5]. Caregivers generally found the tip sheets' format to be inviting, accessible and easy to understand; however, some challenges were identified including the online tip sheet format and a lack of tips focused on younger children. In addition, caregivers noted that some of their personal and cultural caregiving norms differed from practices out-lined in the sheets [5].

To our knowledge, our study, Champions of Positive Parenting 4 Kids (*CHAMPP4KIDS*), will be the first study to assess the acceptability and feasibility of the PLH parenting tip sheets and social media ads in a Canadian context. Results from our study will inform future use of the PLH parenting resources beyond the COVID-19 pandemic and in different contexts, while also contributing to the limited evidence base on the feasibility and acceptability of mass media parenting interventions generally. *CHAMPP4KIDS* study participants will include both early childhood service providers across multiple sectors and caregivers of children aged two to six years in Ontario, Canada. We are especially interested in understanding if the tip sheets are acceptable and feasible for caregivers who are racialized, marginalized and/or typically face barriers to accessing mental health services and parenting programs.

Our explorations of the acceptability of the PLH materials and the feasibility of the materi-als being adopted by service providers and caregivers in a Canadian context are informed by constructs outlined in Rogers' Diffusion of Innovation (DOI) Theory [64] and the Consoli-dated Framework for Implementation Research Constructs (CFIRC) [65]. The DOI and the CFIRC contend that the constructs such as complexity, compatibility, trialability, and design of an innovation may facilitate its acceptability and feasibility [64, 65]. To understand the materials' acceptability, we will first gather service providers' and caregivers' impressions of the tip sheets and social media ads including the comprehensiveness and clarity of the PLH materials and whether they are visually appealing, engaging, relevant and compatible with dif-ferent populations' values [64, 65]. We will then examine the feasibility of service providers and caregivers adopting the materials for use in their professional (service providers) or per-sonal (caregivers) lives. We will specifically explore the complexity of the materials, their trial-ability, relevance and adoption [64, 65].

## Methods

The *CHAMPP4KIDS* study will use a convergent mixed-methods design [66] to assess the acceptability and feasibility of the PLH parenting tip sheets and social media ads from the per-spectives of early childhood service providers and caregivers of children aged 2–6 years in

Ontario, Canada. Part of this study includes a more in-depth exploration of the specific experiences of caregivers who identify with a racialized group, are newcomers or immigrants and/or face barriers to accessing parenting programs.

To address the acceptability of the PLH materials from the perspective of service providers working with families of young children and caregivers of young children (ages 2–6 years), we aim to address the following questions:

1. Are the materials compatible with service providers' professional working environments and the contexts of the populations they serve?

2. Are materials compatible with caregivers' values, perspectives and parenting realities?

3. Are the materials understandable, engaging, comprehensive and useful?

4. How do providers' and caregivers' perspectives on the materials explain their perceived acceptability of the materials?

5. How do providers' and caregivers' perspectives on the acceptability of the materials compare to each other?

To address the feasibility of the PLH materials being adopted and used by service providers working with families of young children and caregivers of young children within a Canadian context, we aim to address the following questions:

1. How will these materials be used?

2. What does one learn from the materials?

3. What barriers are there to using the materials?

4. How best can the materials be delivered for maximum impact and use?

5. Are caregivers' scores on mood associated with survey responses to the PLH tip sheets and social media ads?

6. How do providers' and caregivers' perspectives on the materials explain their perceived feasibility of the materials?

7. How do providers' and caregivers' perspectives on feasibility compare with each other?

## Recruitment and sampling

**Service provider recruitment and sampling.** Early childhood service providers (n = 200), including managers, supervisors, and direct service providers, working with families with children aged 2–6 years in Ontario, Canada will be recruited from community organizations across various sectors. Providers are defined as individuals who provide mental health services and parenting programs, or other supports working within community organizations such as daycare centres, adult education, and settlement services for newcomer families. The Research Coordinator will send a standardized email to community partners which includes a Hamilton Integrated Research Ethics Board (HiREB) approved flyer advertising the study and an invitation to participate. Once a positive response is received, the Research Coordinator will email a link directing providers to a (HiREB) informed consent form (ICF). Targeted organizations include public health units across Ontario responsible for delivering the provincial *Healthy Babies, Healthy Children* program, mental health agencies, managers and service providers of *EarlyON* child and family centres and other early childhood organizations.

After consenting to participate in the study, the provider will be directed to the online survey. At the end of the survey, providers will be asked if they are willing to participate in a focus group discussion to gather additional information and insight on the PLH tip sheets, social media ads and the populations they serve. The Research Coordinator will contact providers who express interest in participating in a focus group (n = up to 40 providers, about five participants per focus group discussion) to obtain a separate ICF and schedule the discussion.

**Caregiver quantitative recruitment and sampling.**   Caregivers (n = 100) will be recruited for the quantitative survey via advertisements posted at community organization service sites, websites. A landing page has been developed to promote the caregiver survey (https://www. champp4kids.com/caregivers). Caregivers who are interested in participating can link to an anonymous screener from the landing page to assess their eligibility for the study survey.

Caregivers will be eligible to participate in the quantitative survey if they:

1. Are the custodial caregiver, living in Canada, with a child aged 2–6 years at the time of study recruitment;

2. Are capable of giving informed, written consent;

3. Have sufficient knowledge of English to complete assessment measures;

4. Identify as a member of a racialized group, a newcomer or an immigrant **or** have not accessed any parenting programs within the last six years.

During the screener, caregivers will be asked to indicate where they heard about the study. If they respond correctly by providing the name of a participating community organization, they will be considered verified and can complete the consent form and survey.

Caregivers who do not meet eligibility criteria via the online screening tool will be redirected to a screen thanking them for their interest in our study. Caregivers meeting the eligibility criteria will be linked to the ICF in RedCap (electronic data capture tool hosted at McMaster University) [67] to consent to study participation via electronic signature. Consenting caregivers will be provided with an electronic copy of their consent form for their records. Once consent is signed, caregivers will be provided with an electronic link through RedCap to the caregiver survey.

**Caregiver qualitative recruitment and sampling.**   Five focus groups of up to 5 caregivers (n = 25) will be used to explore the experiences of marginalized caregivers to better understand the disparities in access to parenting programs which exist between different caregiver groups.

Caregivers will be purposively selected for focus groups if they:

1. Are the custodial caregiver of a child aged 2–6 years at the time of study recruitment;

2. Are capable of giving informed, written consent;

3. Are able to converse and comprehend questions posed in English;

4. Identify as a member of a racialized group, a newcomer or an immigrant **or** have not accessed any parenting programs within the last six years.

Recruitment for qualitative sampling will be conducted through our existing partnerships with organizations that support newcomer, immigrant, and racialized families in the Hamilton, Ontario, Canada area. The Research Coordinator will work in conjunction with local organizations to help promote the study using HiREB-approved social media ads, identify potential participants and conduct the informed consent process either via phone or in person at the organization's office. Consenting caregivers will be informed that they are being asked to participate in two focus groups (see below). If they agree, they will be scheduled for the first

focus group. This study was approved by the Hamilton Integrated Research Ethics Board (HiREB Project # 15065). Informed consent will be obtained from all participants prior to participation.

## Study design

Quantitative surveys will be administered online. Focus groups with providers will be held online, while focus groups with caregivers will take place in person in the community at partnering organizations. Partnering organizations have provided letters of support documenting their agreement to host the focus groups. If in-person focus groups are not possible or cause an undue amount of inconvenience to caregivers, they will be held online. Focus groups will be facilitated in English by a trained and experienced qualitative researcher and will be audio recorded with participants' permission.

**Service provider quantitative survey.** Service providers who express interest in participating in the quantitative survey will receive an email containing digital copies of the tip sheets and social media ads and a link to the online ICF and the survey. Consenting participants will be directed to the survey, beginning with demographic questions. Then, providers will be asked to indicate their level of agreement with various statements related to the tip sheets using a Likert Scale (1 = *'strongly disagree'* to 5 = *'strongly agree'*). These statements were informed by attributes of innovation which can affect behaviour change from Rogers' Diffusion of Innovation Theory [64] and the CFIRC (see Table 1) [64, 65]. Providers will be able to

**Table 1. Ranking attributes of innovation which can affect uptake, provider survey [64, 65].**

**Adoption of materials**

 1. *These materials will educate my clients by helping them understand various tips and techniques.*

 2. *These tip sheets are effective in delivering their messages on the stated topic.*

**Compatibility**

 3. *The items and information in these materials align with your beliefs and values.*

 4. *These materials will benefit clients/ caregivers in our practice.*

 5. *Parents and caregivers will feel confident enough to utilize the tips mentioned in the materials.*

 6. *The materials use language that is condescending or patronizing.*

 7. *The materials use images that are condescending or patronizing.*

 8. *Parents and caregivers have enough information provided on the tip sheets to put these strategies into practice.*

**Complexity**

 9. *The materials are easy to read and use.*

 10. *The materials are clear and understandable.*

**Trialability**

 11. *I could use these materials in my practice, and they would be easily adopted by my clients.*

 12. *I could modify the materials to suit the goals of my practice.*

**Observability**

 13. *These materials would be beneficial to my clients/ caregivers.*

**Visual Appeal**

 14. *The visual aids in the materials encompass all the information presented in them.*

 15. *The visual layout of the materials makes them easy to read and understand.*

 16. *I have a general understanding of what will be discussed in the materials based on their infographics.*

**Understandability**

 17. *The information presented in the materials is easy to understand.*

**Readability**

 18. *I can easily read the information presented in the materials.*

access to all relevant PLH materials (tip sheets and social media ads) as needed during the survey.

Providers will then be asked to rank the eight tip sheets from the *'most important'* for the families with whom they work to the *'least important'*. For each of the eight tip sheets, providers will also be asked to indicate if they *'would use'* or *'would not use'* the sheet in their practice, with the option to select *'prefer not to answer'*. Providers will have the opportunity to respond to open-ended questions to indicate if they feel any topics were missing from the tip sheets, suggest topics which would respond to the needs of the populations they serve and offer comments specific to any of the tip sheets individually. After reflecting on the tip sheets, providers will repeat the *'most important'* to *'least important'* ranking exercise described above for seven social media ads also from the PLH material package. Then, they will have the chance to comment and offer feedback on any or all of the social media ads individually.

RedCap software will notify the research team after each survey completion using a survey identification number. Upon survey completion, an electronic gift card will be issued to the participant's email address as a thank you for their time. Participants' personal information will be stored on a secure file and survey responses will be kept anonymous and confidential.

**Service provider focus groups.** Providers who indicate interest in participating in a focus group at the end of the survey will be contacted by the Research Coordinator to complete a separate ICF and schedule a discussion time. These providers will again be sent electronic versions of each of the social media ads and tip sheets to review before the focus group. Focus groups will begin with two interactive warm-up activities. First, each provider will be asked to select the top three PLH social media ads they think would have the most impact on caregivers of young children. After discussing the reasoning behind their selections, providers will be asked to choose the top three tip sheets they think address some of the biggest challenges facing families they serve, and discuss their selections. These exercises will lead into a semi-structured discussion on the acceptability of the PLH materials from the providers' perspectives, including the format, presentation, cultural adaptation, responsiveness and utility of parenting information communicated on the sheets. Then, the group will reflect on the feasibility of incorporating the materials into providers' interactions with families and the scale up of mass media distribution.

Focus groups will be limited to 90 minutes. After the focus group is complete, service providers will receive an electronic gift card via email as a thank you for their time.

**Caregiver quantitative survey.** The online caregiver survey will take approximately 30–45 minutes to complete, beginning with demographic questions. Then, as with providers, caregivers will be asked to rate the tip sheets on various attributes of innovation using a Likert scale (1 = *'strongly disagree'* to 5 = *'strongly agree'*) (Table 2) [64, 65]. Caregivers will be able to access all relevant PLH materials during the survey as needed. The language used in the Caregiver survey has been formulated to accommodate a grade 8 literacy level to ensure equal access to all participants (https://goodcalculators.com/flesch-kincaid-calculator/).

After rating each of the eight tip sheets on the criteria above, caregivers will be asked to rank the eight tip sheets from the *'most important'* to the *'least important'* for their family. Then, caregivers will be asked to indicate if they *'would use'* or *'would not use'* each of the tip sheets with their child, with the option to select *'prefer not to answer'* for any of the sheets. Caregivers will have the opportunity to comment on the tip sheets individually and suggest topics which they felt were important for their families but missing from the materials. Caregivers will then be asked to reflect on social media ads from the PLH materials. First, they will repeat the *'most important'* to *'least important'* ranking exercise described above for seven social media ads. Then, they will have the chance to comment on the social media ads individually.

**Table 2. Rating attributes of innovation which can affect uptake, caregiver survey [64, 65].**

| **Adoption of materials** |
|---|
| 1. *These materials will help teach me and my family by providing tips I can use in my everyday life.* |
| **Compatibility** |
| 2. *The information in these materials fit with my family's beliefs and values.* |
| 3. *I feel these materials were created for families like mine.* |
| 4. *The materials use language that is bossy.* |
| 5. *The materials use images that are childish.* |
| **Complexity** |
| 6. *The materials are easy to read and use.* |
| 7. *The materials are clear and understandable.* |
| **Trialability** |
| 8. *I could easily use these materials in my home, and they would be accepted well in my family.* |
| 9. *Each sheet suggests at least one, clear action I can practice in my own home with my family.* |
| 10.*I could slightly change the materials to suit the goals and values of my family.* |
| **Visual appeal** |
| 11. *The visual layout of the materials makes them easy to read and understand.* |
| 12. *I have a general understanding of what will be discussed in the materials based on their graphics* |
| 13. *The cartoons used on the materials add something positive to information sheets.* |
| **Understandability** |
| 14. *The information presented in the materials are easy to understand.* |
| **Readability** |
| 15. *I can easily read the information presented in the materials.* |
| 16. *The level of grammar used in the materials is not too complicated.* |
| **Relevance to Your Home** |
| 17. *The tip sheets are relevant to my parenting at home.* |
| **Behavioural Intention** |
| 18. *I would use these materials into my parenting practices.* |
| **Perception of Accountability** |
| 19. *Other caregivers would use these materials.* |

Finally, caregivers will be asked to complete two self-report scales. The first will measure depression called, The Centre for Epidemiologic Studies Depression Scale (CES-D) and the second anxiety The Generalized Anxiety Disorder scale (GAD-7). These two scales are commonly used measures in various parenting program evaluations. As such we will conduct a general comparison to examine if caregiver mental health symptomatology is correlated with their perceptions of acceptability and feasibility of the PLH materials.

The (CES-D) is a structured, 20 item self-report scale designed to measure current levels of depressive symptomatology in the general population [68] and help identify those at risk for clinical depression [69]. The scale asks about symptoms that occurred in the week prior to the survey, with items (frequency of symptoms) rated on a 4-point Likert scale. The scale takes between two and five minutes to complete [70]. The CES-D shows high internal consistency, good sensitivity and specificity, acceptable test-retest stability, excellent concurrent validity by clinical and self-report criteria, and substantial evidence of construct validity [69, 70].

The (GAD-7) is a 7-question scale used to screen for presence and severity of GAD [71]. The GAD-7 takes between one to two minutes to complete [72] and has demonstrated good psychometric properties, including internal consistence and convergent validity with the Beck Anxiety Inventory [71].

RedCap software will notify the research team after each survey completion using a survey identification number. Upon survey completion, an electronic gift card will be issued to the participant's email address as a thank you for their time. Participants' personal information will be stored on a secure file and survey responses will be kept anonymous and confidential.

**Caregiver focus groups.** We will use an adapted *Trials of Improved Practices* (TIPS) methodology [73] to better understand the feasibility of caregiver uptake of PLH parenting practices. TIPS is often used in the formative phases of behaviour change intervention development to "pre-test the actual practices a program will promote" with the intervention's target population, focusing on people's interpretations of practices and actions rather than their existing knowledge or beliefs [73, 74]. TIPS will allow us to explore the acceptability of PLH tip sheets to caregivers and the feasibility of caregivers implementing parenting strategies in the applied, realistic contexts of their own homes. TIPS has been used successfully in other qualitative studies to explore the acceptability and feasibility of early childhood development, parenting recommendations and associated materials, including message dissemination via mass media [73, 74].

The TIPS method requires at least two contacts with participants. In our study, the first round of focus groups with caregivers will begin with an interactive warm-up activity, asking caregivers to select the three tip sheets which they think address some of the biggest challenges facing caregivers of young children. Each participant will be invited to share their selection and justification. This warm-up activity will lead into a semi-structured discussion on caregiving norms, caregivers' self-identified successes and challenges caring for children aged 2–6 years and caregiver perceptions of the effects of stress on caregiver, child and overall household well-being. To ensure focus groups do not last beyond 90 minutes, caregivers will not rank or discuss social media ads.

At the end of the first focus group, the facilitator will display the four tip sheets which were most frequently mentioned by participants during the warm-up activity. The facilitator will share and briefly describe the basic theme and parenting suggestions on each of these tip sheets. Concise, standardized descriptions of each tip sheet will be prepared before the focus group so that participants receive the same orientation across groups. Participants will be asked to read the four tip sheets after the focus group and select at least two sheets to practice at home for the following two weeks. To reduce the risk of social desirability bias influencing caregiver responses [75], the facilitator will emphasize that these activities are not a "test" of the caregiver's ability to implement the parenting behaviours; rather, we are interested in understanding every aspect of caregivers' experiences with the tip sheets, even if they are not able to implement any of the suggestions or find the suggested behaviours difficult or irrelevant.

Before the first focus group concludes, the facilitator will ask for each participant's verbal permission to call them on their personal phone on a pre-determined day. Tip sheets will be given to participants as hard copies (in person focus groups) or sent again to the participants via email (virtual focus groups) for them to reference at home during the TIPS. Participants will be reminded that a second focus group discussion will be held in two weeks' time to discuss their experiences with the tip sheets and PLH suggested parenting practices. The facilitator will again emphasize that we will be interested in hearing *all* experiences with the tip sheets even if participants are not able to implement any behaviours described on the sheets.

About four days after the first focus group, the facilitator will call each caregiver to "check in" and conduct a short, semi-structured qualitative interview to better understand caregiver perceptions of and reactions to the tip sheets and suggested behaviours, the feasibility of practicing the behaviours with their own children and any successes or challenges they have faced in the first days with the tip sheets. The facilitator will remind the participant of the next focus

group, and again encourage them to attend even if they are not able to or do not want to implement the suggested parenting practices.

The second focus group will take place two weeks after the first focus group. Caregivers will be asked to share their experiences with using the tip sheets in their own homes. The facilitator will ask caregivers to describe the behaviours they chose to practice with their families, frequency of practice, successes and challenges to implementing the behaviours, their own reactions to implementing the behaviours, child and other family members' reactions to implementing the behaviours and the overall relevance of the behaviours to the caregivers' lives and their families' needs.

## Study outcomes & analysis

The primary outcomes of the quantitative surveys will be analyzed by a research team member and include participant demographics and descriptive statistics (means, standard deviations and proportions) of Likert Scale scores given by providers and caregivers when rating various aspects of the tip sheets (see Tables 1 and 2). For caregivers, descriptive analysis will include responses to the CES-D and GAD-7. All quantitative analyses will be performed using SPSS v.28.

Qualitative data will be analyzed using the Rapid Qualitative Inquiry (RQI) method. RQI is a team-based, applied research method designed for implementation and policy research to quickly develop an inside perspective on and preliminary understanding of complex "on-the-ground" situations [76]. This rapid, "not rushed" approach uses a reflexive, iterative, team-focused data analysis that provides timely and actionable results. Qualitative data will be analyzed by two team members trained in the RQI process. Initially, provider and caregiver data will be analyzed separately. The RQI process will be repeated for each participant group. Once analysis is complete we will construct a convergence coding matrix to triangulate provider and caregiver perceptions of acceptability and feasibility of the tip sheets [77]. All interview recordings will be transcribed verbatim with identifying information removed by a professional transcription service. Two analysts, with public health or health services research expertise, will engage in the analysis process. The first step will be the development of a draft summary template document for each unique data source and type. The two researchers will read two transcripts in their entirety and use the draft template for data extraction while revising the template to ensure that all content domains were included. The remaining transcripts will be divided between the two analysts. To promote dependability in data extraction, two transcripts will be reviewed and synthesized by the analysts. Once consistency in data extraction and synthesis had been established across both analysts, the remaining transcripts will be individually summarized within the domain templates. From the templates, data synthesized for each domain will then be collated by domain within an Excel spreadsheet. Domains will then be assigned to each analyst, and key findings will be synthesized for each data source. Analysts will schedule consistent meetings to discuss and triangulate findings and conclusions as analysis progresses and ensure the study's main research questions are answered.

To integrate quantitative and qualitative findings, the research team will follow a triangulation protocol [78]. After analysis of quantitative and qualitative data is complete, the research team will work together to construct a convergent joint display table [66]. For each participant group, findings from both methods of data collection will be listed together by theoretical constructs [64, 65] and compared to understand where data converges, is complementary or contradictory and if any data set is "silent", or does not offer insight, on a particular aspect of the research questions [78]. A final stage of analysis will compare the perspectives of providers and caregivers.

The risks to participants in this study are minimal. All data will be kept confidential and anonymous (de-identified using a numerical participant study ID) during analysis and reporting. Only select members of the research team will have access to the raw data and this data will not be shared with any other person or organization. If focus groups are conducted virtually, Zoom Telehealth will be used to record discussions as this virtual platform complies with the Canadian federal privacy law (Personal Information Protection and Electronic Documents Act) and the Ontario privacy law (Personal Health Information Protection Act). There is a small risk of privacy breach for data collected on external servers. If any participants share concerns about this possibility, the research team will make alternative arrangements for participation (ie via telephone). At the beginning of each focus group, all participants will be asked to keep the views of others in the groups confidential. However, there is a risk that participants may not follow this request. A confidentiality breach is unlikely to have a detrimental impact on participants as the interview topics are not sensitive.

Providers and caregivers will be compensated for their time completing questionnaires with a gift card and at the end of the focus groups with a gift card.

## Discussion

Caregiver stress can have a significant and lasting impact on the physical and mental health of both caregivers and their children [79, 80]. The stressors of daily life were exacerbated for caregivers and their families during the height of the COVID-19 pandemic and continue to pose challenges for many families as they cope with the long-term effects of illness, social isolation, financial pressures and other compounded stressors [38, 42, 58]. Evidence shows that parenting programs can improve the health and well-being of both caregivers and children [6, 7, 15], interventions which are needed now more than ever. However, many families face substantial barriers accessing traditional parenting support services, a reality which can heighten the already significant and inequitable challenges facing racialized, immigrant and otherwise marginalized families.

Resources developed by the PLH group in collaboration with WHO/UNICEF to support caregivers and families around the world during the COVID-19 pandemic also have the potential to reach caregivers and children facing barriers accessing traditional services. As a mass-media campaign, the PLH COVID-19 parenting tip sheets and social media ads offer parenting information and concrete strategies to build positive caregiver-child relationships in succinct, engaging formats for a range of child ages. Materials can be easily distributed and accessed without requiring caregivers and families to incur direct and indirect costs such as expenses and time related to travel, childcare or programs delivered through multiple in-person meetings.

However, given the relatively recent development and dissemination of the PLH materials, evidence on their acceptability and feasibility in different contexts, especially with racialized, immigrant or otherwise marginalized caregivers and families, is limited. The research proposed here will provide significant insights into provider and caregiver perspectives on various aspects of the PLH tip sheets and social media ads including those which might affect uptake, such as comprehensiveness, clarity of messages and if the sheets are engaging, useful and relevant to families' lives and needs in a Canadian context. We will also explore the ability of mass media materials to breach barriers many families face to accessing traditional parenting services. A particular focus on the cultural appropriateness of the materials will be especially important to inform cultural adaptation of parenting materials to fit diverse populations and approaches to parenting in Canada and elsewhere [81–83]. Our mixed methods approach will offer a multi-dimensional perspective on the tools themselves and will allow for triangulation

of quantitative and qualitative results, and perspectives of different participant groups. For example, by gathering the perspectives of both providers and caregivers on the comprehensiveness, pertinence, and appropriateness of materials we will be able to comment on the agreement or divergent perspectives between providers and the populations they serve. In addition, the innovative TIPS methodology will provide in-depth perspective on the facilitators and barriers of practical application of the materials as caregivers will have the opportunity to interpret and implement the information in the realistic context of their own homes.

While the results of this study will make important contributions to the evidence base on mass media parenting interventions, the study will also offer insight into using TIPS methodologies to assess the feasibility and acceptability of various early childhood development interventions in different socio-cultural contexts. A growing literature base highlights the Euro/American-centric origins of many early childhood development interventions and materials, and various authors advocate for ensuring that the cultural adaptation of these materials is intentional and informed by the target populations themselves [82–85]. The TIPS methodology could prove a useful tool for participant-led adaptation of existing parenting, early childhood development and other health intervention materials for a variety of contexts.

## Strengths and limitations

Our study design has several strengths. First, our recruitment and participation will benefit from already-established relationships between the research team and relevant community organizations and providers. In addition, the inclusion of both provider and caregiver perspectives using a mixed-methods approach will allow for the triangulation of results, a comparative analysis of provider and caregiver perspectives and expansive understanding of results. The research team will be intentional in validating qualitative results using a structured, team-based, reflexive approach to analysis and will ensure quantitative and qualitative results are understood in conversation with each other. Finally, our quantitative data collection is easily completed virtually so participants can complete surveys at their convenience, from their homes. Qualitative methods are easily adaptable to both online and in-person data collection to respond to any changes in public health recommendations and participant logistics.

We have attempted to pre-emptively address possible study limitations in our study protocol. First, the already high demands on provider and caregiver time could affect our study response and retention rates. Data collection methods and protocols for follow up with participants were developed to overcome these potential challenges. Also, by nature of the method itself, participants in TIPS are easily attuned to behaviours and practices which are seen as "ideal" or "desirable." Therefore, there is a risk of social desirability bias influencing participant responses [75], especially during focus groups which ask participants to discuss their experiences implementing behaviours from the tip sheets in their households. We will attempt to mitigate the potential influence of social desirability bias throughout our explanations to participants of focus groups and overall study objectives, which include understanding how the PLH materials can be improved from caregivers' tangible experiences rather than wanting to "test" caregivers on their ability to implement suggested activities. We did not collect data on child behavioural measurements, such as the Strengths and Difficulties Questionnaire, in the caregiver quantitative survey as this data could link child behaviour to caregiver acceptability and feasibility of the PLH materials. This could be a limitation that the authors were willing to sacrifice to minimize caregiver survey burden. We will consider collecting these types of measurements in our future work. Furthermore, while our study results will have the potential to inform several aspects of child development and parenting interventions in diverse contexts, our study population will represent specific sub-populations within an English-speaking

Canadian context. Therefore, our results will not necessarily be generalizable to other populations such as non-English speaking caregivers within Canada or elsewhere globally.

## Acknowledgments

The authors would like to thank Drs. Jamie Lachman, Lucie Cluver, Catherine Ward, and Frances Gardner for developing the materials and generous support of this study.

## Author Contributions

**Conceptualization:** Andrea Gonzalez, Amanda Sim, Jenna Ratcliffe, Mari Dumbaugh.

**Data curation:** Andrea Gonzalez.

**Formal analysis:** Susan M. Jack, Amanda Sim, Jenna Ratcliffe.

**Funding acquisition:** Andrea Gonzalez, Harriet L. MacMillan.

**Investigation:** Andrea Gonzalez, Harriet L. MacMillan.

**Methodology:** Andrea Gonzalez, Susan M. Jack, Amanda Sim, Jenna Ratcliffe, Mari Dumbaugh.

**Project administration:** Andrea Gonzalez, Jenna Ratcliffe.

**Resources:** Andrea Gonzalez.

**Supervision:** Andrea Gonzalez, Mari Dumbaugh.

**Validation:** Andrea Gonzalez.

**Writing – original draft:** Andrea Gonzalez, Susan M. Jack, Jenna Ratcliffe, Mari Dumbaugh, Teresa Bennett.

**Writing – review & editing:** Andrea Gonzalez, Susan M. Jack, Amanda Sim, Jenna Ratcliffe, Mari Dumbaugh, Teresa Bennett, Harriet L. MacMillan.

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
