## [Decision Letter · Decision Letter 0]

20 Jul 2023

PONE-D-23-13544CHAMPP4KIDS : mixed methods study protocol to evaluate acceptability and feasibility of Parenting for Lifelong Health materials in a Canadian contextPLOS ONE

Dear Dr. Gonzalez,

Thank you for submitting your manuscript to PLOS ONE. After careful consideration, we feel that it has merit but does not fully meet PLOS ONE’s publication criteria as it currently stands. Therefore, we invite you to submit a revised version of the manuscript that addresses the points raised during the review process.

The manuscript has been evaluated by two reviewers, and their comments are available below.

The reviewers have raised some concerns that need attention. They request additional information on methodological aspects of the study, and in clarification about the eligibility criteria and potential limitations.

Could you please revise the manuscript to carefully address the concerns raised?

We look forward to receiving your revised manuscript.

Kind regards,

Marianne Clemence

Staff Editor

PLOS ONE

Reviewers' comments:

Reviewer's Responses to Questions

**Comments to the Author**

1. Does the manuscript provide a valid rationale for the proposed study, with clearly identified and justified research questions?

Reviewer #1: Partly

Reviewer #2: Yes

2. Is the protocol technically sound and planned in a manner that will lead to a meaningful outcome and allow testing the stated hypotheses?

Reviewer #1: Yes

Reviewer #2: Yes

3. Is the methodology feasible and described in sufficient detail to allow the work to be replicable?

Reviewer #1: Yes

Reviewer #2: Yes

4. Have the authors described where all data underlying the findings will be made available when the study is complete?

Reviewer #1: No

Reviewer #2: No

5. Is the manuscript presented in an intelligible fashion and written in standard English?

Reviewer #1: Yes

Reviewer #2: Yes

6. Review Comments to the Author

You may also provide optional suggestions and comments to authors that they might find helpful in planning their study.

Reviewer #1: Thank you for the opportunity to review this protocol of an important and rigorous study. Overall, I thought the manuscript was clear and organized. There are a couple of areas that I believe would strengthen the manuscript:

1. p.8 paragraph 2 the authors introduce the constructs that will be assessed to understand acceptability. It would be useful for a bit more information related to how these constructs are defined as this would be helpful in connecting the constructs to the proposed measures.

2. p. 11 - the authors describe the eligibility crieria - it would be helpful if there was a description of how eligibility will be determined and verified. Particularly because there doesn't seem to be direct contact with participants and study staff until the focus groups. Also, I'm assuming that participants need to be living in Canada, however, I'm not sure if that is an accurate assumption. Therefore, clarification here would be useful.

3. Rationale for why the study is colleccting CES-D and GAD-7 is needed and how this information informs the acceptability and feasibility evaluation

Reviewer #2: The study is well designed for the objectives pursued, however I have two concerns that the authors need to consider:

- parents to be eligible have to be proficient in English (which may exclude parents from more vulnerable groups and by feeling excluded and without another answer this may contribute to increase their exclusion);

- not asking parents to fill in a rating scale of their child's behavioral problems , as these parents are the ones who get the most out of the programs as they are more motivated.

7. PLOS authors have the option to publish the peer review history of their article (what does this mean?). If published, this will include your full peer review and any attached files.

Reviewer #1: No

Reviewer #2: **Yes: **Maria Filomena Ribeiro da Fonseca Gaspar

---

## [Author Response · Author response to Decision Letter 0]

18 Aug 2023

We thank the editorial team for their feedback and guidance on formatting issues. We have reformatted the manuscript to conform to PLOS ONE’s style requirements. 

We have updated all of the DOIs in the reference section of our manuscript. 

We have deleted the ethics statements from all other sections of the manuscript except in the Methods. 

Reviewers' comments:

Reviewer's Responses to Questions

Comments to the Author

1. Does the manuscript provide a valid rationale for the proposed study, with clearly identified and justified research questions?

Reviewer #1: Partly

Reviewer #2: Yes

2. Is the protocol technically sound and planned in a manner that will lead to a meaningful outcome and allow testing the stated hypotheses?

Reviewer #1: Yes

Reviewer #2: Yes

3. Is the methodology feasible and described in sufficient detail to allow the work to be replicable?

Reviewer #1: Yes

Reviewer #2: Yes

4. Have the authors described where all data underlying the findings will be made available when the study is complete?

The PLOS Data policy requires authors to make all data underlying the findings described in their manuscript fully available without restriction, with rare exception, at the time of publication. The data should be provided as part of the manuscript or its supporting information or deposited to a public repository. For example, in addition to summary statistics, the data points behind means, medians and variance measures should be available. If there are restrictions on publicly sharing data—e.g., participant privacy or use of data from a third party—those must be specified.

Reviewer #1: No

Reviewer #2: No 

Because this is a protocol paper, without any data, we selected ‘N/A – No results are reported’ option under data sharing. Please let us know if this is incorrect and we will rectify the situation. Once the project is completed, we will be happy to share data upon request. 

5. Is the manuscript presented in an intelligible fashion and written in standard English?

Reviewer #1: Yes

Reviewer #2: Yes

6. Review Comments to the Author

You may also provide optional suggestions and comments to authors that they might find helpful in planning their study.

Reviewer #1: Thank you for the opportunity to review this protocol of an important and rigorous study. Overall, I thought the manuscript was clear and organized. There are a couple of areas that I believe would strengthen the manuscript:

We thank the reviewer for their thoughtful and thorough review which has strengthened the clarity of the paper. We hope our edits address their concerns adequately. 

1. p.8 paragraph 2 the authors introduce the constructs that will be assessed to understand acceptability. It would be useful for a bit more information related to how these constructs are defined as this would be helpful in connecting the constructs to the proposed measures.

Please see lines 248-250which now read: The DOI (64) and the CFIRC(65)contend that the constructs such as complexity, compatibility, trialability, and design of an innovation may facilitate its acceptability and feasibility of an innovation (64, 65). Also: we added DOI to the list of abbreviations.

We have also added definitions to Table 1. Please see line 376 Table 1 for construct definitions 

Adoption of Materials: The likelihood that decision makers will decide to put the innovation in place (65).

Compatibility: The degree to which the innovation is perceived to be consistent with socio-cultural values, previous ideas, and/or perceived needs (64).

Complexity The degree to which an innovation is difficult to use or understand, its simplicity (64).

Trialability: The ability to test the intervention on a small scale in the organization and to be able to reverse course if needed (65).

Observability: The degree to which the results of an innovation are visible to potential adopters (64).

Visual Appeal: the degree to which the innovation is well designed and presented (65).

Understandability: the degree in which the innovation is seen as difficult to understand or use (64).

Readability: the degree in which the innovation is seen as difficult to read or comprehend (64).

2. p. 11 - the authors describe the eligibility criteria - it would be helpful if there was a description of how eligibility will be determined and verified. Particularly because there doesn't seem to be direct contact with participants and study staff until the focus groups. Also, I'm assuming that participants need to be living in Canada, however, I'm not sure if that is an accurate assumption. Therefore, clarification here would be useful.

Thank you for this point of clarification. We attempt to verify participant eligibility in two ways. First, for the focus groups, we have established partnerships with various community organizations, and they are sharing our recruitment flyers within relevant groups, settings, and directly with clients and members that attend their various programs. Thus, participants who are approached about the study generally meet the eligibility criteria. Eligibility criteria is further confirmed by the Research Assistant, during the consent call using the four pre-screener questions used in the quantitative survey, explained further. Second, for the quantitative survey, we have a pre-screener in place which involves responding to four questions, including: Do you have a child between the ages of 2-6 years old? Y Do you identify as a member of a racialized group? Are you a newcomer or immigrant to Canada? 

Please see the following edited sections:

Line 323 (tracked changes copy): Are the custodial caregiver, living in Canada, with a child aged 2-6 years at the time of study recruitment. And lines 329-332: listing the 4 screener questions: During the screener, the following questions will be presented to participants: Do you have a child between the ages of 2-6 years old? Have you participated in a parenting program such as Triple P, Nobody's Perfect or Circle of Security within the last 6 years? Do you identify as a member of a racialized group? Are you a newcomer or immigrant to Canada?

3. Rationale for why the study is collecting CES-D and GAD-7 is needed and how this information informs the acceptability and feasibility evaluation.

Please see Line 438-444 (tracked changes copy): Finally, caregivers will be asked to complete two self-report scales. The first will measure depression called, The Centre for Epidemiologic Studies Depression Scale (CES-D) and the second anxiety The Generalized Anxiety Disorder scale (GAD-7). These two scales are commonly used measures in various parenting program evaluations. As such we will conduct a general comparison to examine if caregiver mental health symptomatology is similar to what is reported in the literature, and whether scores correlated with their perceptions of acceptability and feasibility of the PLH materials. 

Reviewer #2: The study is well designed for the objectives pursued, however I have two concerns that the authors need to consider. 

We thank the reviewer for their positive feedback and their concerns outlined below. We provided additional information regarding the recruitment, eligibility criteria and why we have not included a measure of child behavioural problems. 

- parents to be eligible have to be proficient in English (which may exclude parents from more vulnerable groups and by feeling excluded and without another answer this may contribute to increase their exclusion); 

We have partnered with various agencies in high priority neighbourhoods to promote advertisement for the study. In addition, several of our partners include Immigration Centres or centres for refugee families. Although the participants still need to speak and read in English, we expect that we will be able to recruit a number of caregivers who represent more vulnerable groups. To address and acknowledge this as a potential limitation, we have added this to our limitation section as follows: Lines 668-670 (tracked changes version): our study population will represent specific sub-populations within an English-speaking Canadian context. Therefore, our results will not necessarily be generalizable to other populations such as non-English speaking caregivers within Canada or elsewhere globally. 

– not asking parents to fill in a rating scale of their child's behavioral problems, as these parents are the ones who get the most out of the programs as they are more motivated.

Although we agree with the reviewer, given we are not currently evaluating a program and that we are attempting to limit the length of the survey, we have not included a child behaviour measure. We have added this as a limitation to the study. Please see Lines 662-665: Although measures of child behaviour problems are typically collected in parenting program evaluations, to minimize participant burden, we did not include such a measure in the current study. Future research examining the effectiveness of the PLH materials should include a range of relevant outcomes, including child functioning.

---

## [Decision Letter · Decision Letter 1]

12 Jan 2024

PONE-D-23-13544R1CHAMPP4KIDS : mixed methods study protocol to evaluate acceptability and feasibility of Parenting for Lifelong Health materials in a Canadian contextPLOS ONE

Dear Dr. Gonzalez,

Thank you for submitting your manuscript to PLOS ONE. After careful consideration, we feel that it has merit but does not fully meet PLOS ONE’s publication criteria as it currently stands. Therefore, we invite you to submit a revised version of the manuscript that addresses the points raised during the review process. Please submit your revised manuscript by Feb 26 2024 11:59PM. If you will need more time than this to complete your revisions, please reply to this message or contact the journal office at plosone@plos.org. Please include the following items when submitting your revised manuscript:A rebuttal letter that responds to each point raised by the academic editor and reviewer(s). You should upload this letter as a separate file labeled 'Response to Reviewers'.A marked-up copy of your manuscript that highlights changes made to the original version. You should upload this as a separate file labeled 'Revised Manuscript with Track Changes'.An unmarked version of your revised paper without tracked changes. You should upload this as a separate file labeled 'Manuscript'.If applicable, we recommend that you deposit your laboratory protocols in protocols.io to enhance the reproducibility of your results. Protocols.io assigns your protocol its own identifier (DOI) so that it can be cited independently in the future. For instructions see: https://journals.plos.org/plosone/s/submission-guidelines#loc-laboratory-protocols. Additionally, PLOS ONE offers an option for publishing peer-reviewed Lab Protocol articles, which describe protocols hosted on protocols.io. Read more information on sharing protocols at https://plos.org/protocols?utm_medium=editorial-email&utm_source=authorletters&utm_campaign=protocols.

We look forward to receiving your revised manuscript.

Kind regards,

Ali Montazeri

Academic Editor

PLOS ONE

Journal Requirements:

Reviewers' comments:

Reviewer's Responses to Questions

**Comments to the Author**

1. Does the manuscript provide a valid rationale for the proposed study, with clearly identified and justified research questions?

Reviewer #1: Yes

Reviewer #2: Yes

2. Is the protocol technically sound and planned in a manner that will lead to a meaningful outcome and allow testing the stated hypotheses?

Reviewer #1: Yes

Reviewer #2: Yes

3. Is the methodology feasible and described in sufficient detail to allow the work to be replicable?

Reviewer #1: Yes

Reviewer #2: Yes

4. Have the authors described where all data underlying the findings will be made available when the study is complete?

Reviewer #1: Yes

Reviewer #2: No

5. Is the manuscript presented in an intelligible fashion and written in standard English?

Reviewer #1: Yes

Reviewer #2: Yes

6. Review Comments to the Author

You may also provide optional suggestions and comments to authors that they might find helpful in planning their study.

Reviewer #1: The authors addressed all comments and I believe the protocol manuscript has been strengthened. This is minor, but the authors may consider how potential participants might interpret the question "are you a member of a racialized group?" - I work with a similar population and I believe parents may not fully understand the question. Therefore, it might be helpful to have some prompts for the RAs. Finally, although I understand the rationale for not including a child behavior measure due to length of survey, I wonder if the investigators considered the Strengths and Difficulties QUestionnaire - it's only 25 items and takes less than 5 minutes to complete and may be a good addition.

Reviewer #2: p1 Line 26: Frances Gardner should replace Frances Gardiner

p.3 line 88 The link for the reference nº14 isn'r working

7. PLOS authors have the option to publish the peer review history of their article (what does this mean?). If published, this will include your full peer review and any attached files.

Reviewer #1: No

Reviewer #2: **Yes: **Maria Filomena Ribeiro da Fonseca Gaspar

---

## [Author Response · Author response to Decision Letter 1]

16 Jan 2024

Response to Reviewers

1. Does the manuscript provide a valid rationale for the proposed study, with clearly identified and justified research questions?

Reviewer #1: Yes

Reviewer #2: Yes________________________________________

2. Is the protocol technically sound and planned in a manner that will lead to a meaningful outcome and allow testing the stated hypotheses?

Reviewer #1: Yes

Reviewer #2: Yes

3. Is the methodology feasible and described in sufficient detail to allow the work to be replicable?

Reviewer #1: Yes

Reviewer #2: Yes

4. Have the authors described where all data underlying the findings will be made available when the study is complete?

Reviewer #1: Yes

Reviewer #2: No

Thank you for this comment. A data availability statement has now been added to the manuscript – please see lines 643-645. 

5. Is the manuscript presented in an intelligible fashion and written in standard English?

Reviewer #1: Yes

Reviewer #2: Yes

6. Review Comments to the Author

You may also provide optional suggestions and comments to authors that they might find helpful in planning their study.

Reviewer #1: The authors addressed all comments and I believe the protocol manuscript has been strengthened. This is minor, but the authors may consider how potential participants might interpret the question "are you a member of a racialized group?" - I work with a similar population and I believe parents may not fully understand the question. Therefore, it might be helpful to have some prompts for the RAs. Finally, although I understand the rationale for not including a child behavior measure due to length of survey, I wonder if the investigators considered the Strengths and Difficulties Questionnaire - it's only 25 items and takes less than 5 minutes to complete and may be a good addition.

We appreciate your suggestion to include prompts for the question "are you a member of a racialized group?" on our online screening page https://www.champp4kids.com/caregivers. While we recognize the potential for ambiguity, our decision not to modify the screener further stems from a balance between ensuring clarity and minimizing the risk of introducing leading cues. We believe that our current approach maintains a level of neutrality and allows participants to self-identify in a manner they find most fitting. However, your feedback will be considered for future refinements as we continually strive to enhance the quality of our research.

Also, thank you for your insightful comment regarding the omission of a child behavior measure due to survey length constraints. We appreciate your suggestion of the Strengths and Difficulties Questionnaire. While we did not include it in the current study, we acknowledge its potential value. Your input will be noted as a limitation in the protocol manuscript, and we will consider it for future directions in our research.

Reviewer #2: p1 Line 26: Frances Gardner should replace Frances Gardiner

p.3 line 88 The link for the reference nº14 isn’t working

Thank you for this edit. The correction to this professor’s name has been corrected. 

Thank you for this comment. We have updated the link in the reference section, so it is searchable and functional.

7. PLOS authors have the option to publish the peer review history of their article (what does this mean?). If published, this will include your full peer review and any attached files.

Do you want your identity to be public for this peer review? For information about this choice, including consent withdrawal, please see our Privacy Policy.

Reviewer #1: No

Reviewer #2: Yes: Maria Filomena Ribeiro da Fonseca Gaspar

---

## [Editor Report · Decision Letter 2]

22 Jan 2024

CHAMPP4KIDS : mixed methods study protocol to evaluate acceptability and feasibility of Parenting for Lifelong Health materials in a Canadian context

PONE-D-23-13544R2

Dear Dr. Gonzalez,

We’re pleased to inform you that your manuscript has been judged scientifically suitable for publication and will be formally accepted for publication once it meets all outstanding technical requirements.

Kind regards,

Ali Montazeri

Academic Editor

PLOS ONE
---

## [Editor Report · Acceptance letter]

28 Feb 2024

PONE-D-23-13544R2 

PLOS ONE

Dear Dr. Gonzalez, 

I'm pleased to inform you that your manuscript has been deemed suitable for publication in PLOS ONE. Congratulations! Your manuscript is now being handed over to our production team.

Kind regards, 

on behalf of

Professor Ali Montazeri 

Academic Editor

PLOS ONE